# Measuring the Effect of Circular Public Procurement on Government's Environmental Impact

**Michiel Zijp** [1,*], **Erik Dekker** [1], **Mara Hauck** [2], **Arjan De Koning** [3], **Marijn Bijleveld** [4], **Janot Tokaya** [2], **Elias De Valk** [1], **Anne Hollander** [1] and **Leo Posthuma** [1,5]

[1] Dutch National Institute for Public Health and the Environment (RIVM), Antonie van Leeuwenhoeklaan 9, 3721 MA Bilthoven, The Netherlands
[2] Netherlands Organization for Applied Scientific Research (TNO), Location Utrecht, Princetonlaan 6, 3584 CB Utrecht, The Netherlands
[3] Institute of Environmental Sciences (CML), Leiden University, 2300 RA Leiden, The Netherlands
[4] CE Delft, Oude Delft 180, 2611 HH Delft, The Netherlands
[5] Department of Environmental Science, Radboud University Nijmegen, 6525 XZ Nijmegen, The Netherlands
[*] Correspondence: michiel.zijp@rivm.nl

**Abstract:** Governments contribute to the transition toward a circular economy (CE) by using criteria in their procurement processes that trigger the supply of circular products and services, namely circular public procurement (CPP). CPP practices are emerging in Europe. However, the effect of CPP is not yet monitored and hence remains unclear. What is the efficacy of CPP in reducing the impacts of goods and services? Analyzing CPP efficacy is an important next step in exploring how to improve its application. This paper presents the results of an effect evaluation of CPP in the Netherlands, using a sample-based mixed-method approach in combination with life cycle assessment for analyzing CPP-induced reduced impacts on global warming and material use. Two thirds of the procurement tenders in which circular procurement criteria were applied in 2017 and 2018 did not result in reduced environmental impacts or reduced material use. One third, however, showed that, as well as how CPP can contribute to the transition toward a CE. The identified remaining challenges are (1) to apply criteria that are ambitious enough to challenge the market and (2) to keep attention on the circularity ambitions up during the implementation phase of the procurement process. Effect indicators are proposed to complement the current monitoring practices of CPP implementation.

**Keywords:** circular public procurement; environmental impact; monitoring; circular economy; indicators; effect





## 1. Introduction

In 2015, 17 Sustainable Development Goals (SDG) were adopted [1]. They are derived from the idea that, in a business-as-usual scenario, more and more environmental and social carrying capacity limits at different spatial scales will be exceeded, and resources will be depleted [2,3]. One of the goals to counter unsustainable trends is to "ensure sustainable consumption and production patterns" (SDG goal 12). Governments can not only influence consumption and production patterns via regulation and legislation but also by their own actions. Government spending is a sizeable part of the total spending on consumer goods and services and therefore has a potentially significant influence on sustainability. In this paper, we evaluate the effects of attempts to steer toward a more sustainable society via government spending.

In the Netherlands alone, the government spent about EUR 85 billion on public procurement in 2019, which is approximately 11% of its gross domestic product (GDP) [4]. For Europe as whole, this adds up to around EUR 2 trillion and almost 14% of its GDP [5]. The size of this expenditure naturally produces opportunities to create an impact and economic driver in future developments. With sustainable public procurement (SPP), governments

make efforts to include external costs and benefits in the selection procedure of the procurement of products and services instead of only the monetary costs and function [6,7]. Such externalities can be environmental or socioeconomic, such as greenhouse gas emissions during the life cycle of products and the inclusion of people with a disadvantage within the job market, respectively. When SPP is only directed at reducing environmental impact, it is generally referred to as green public procurement (GPP) [8]. Through SPP's governmental "purchasing power", focused government implementation is thought to be able to boost a society-wide trend to develop and demand for sustainable products. The idea is that an increase in demand allows suppliers to invest in more sustainable production processes and, as a consequence, increases their supply to the benefit of other purchasers. As such, SPP is thought to contribute both directly and indirectly toward the SDG policy goals, which is the reason that in the past decennium, many governments have pursued its implementation [9].

In line with the increasing attention for the transition toward a circular economy (CE) in both policy and research, recently, circular public procurement (CPP) has been introduced as a new form of SPP [10]. CPP has its own specific approach to sustainability and has been defined as "the process by which public authorities purchase works, goods or services that seek to contribute to closed energy and material loops within supply chains, whilst minimizing, and in the best case avoiding, negative environmental impacts and waste creation across their whole life-cycle" (p. 5 [11]). Although many different definitions of CE exist [12], its general goal is to change our predominantly linear use of materials to a circular mode of production and consumption, where in the former, materials and products are mostly wasted after having been used once, and in a CE, they are reused and recycled to lower the amount of waste and the use of virgin materials [13,14]. A transition to a CE is expected to contribute to reducing environmental impacts, to secure resources and raw materials and to create opportunities for novel policy approaches and business models. CPP is expected to contribute to this transition. Both Alhola et al. (2018) and Sönnichsen et al. (2020) emphasized the similarities between SPP, GPP and CPP [10,15]. Based on CPP practices, Alhola et al. (2018) distinguished four ways in which contracting authorities implement CPP (i.e., by "the procurement of better-quality products in circular terms, the procurement of new circular products, the use of business concepts that support the CE and investments in circular ecosystems") [15]. They argue that the main overlaps of CPP with SPP and GPP are in the first two "product- and technology-oriented" approaches and that CPP adds the circular system perspective of the latter two.

Research on the implementation of SPP and GPP has increased in the last few years. Studies from different regions in the world, both in developed and developing countries, investigated the conditions that support or hamper SPP implementation using questionnaires, mainly Walker and Brammer (2009, UK), Large and Gimenez Thomsen (2011, Germany), Roman (2017, US), Zaidi et al. (2019, Pakistan), Ye et al. (2021, China) and Kannan (2021, Denmark) [16–20]. Research on the effect of SPP and CPP, however, is very limited [21]. This is problematic because attention on circularity or sustainability during procurement and even the application of CPP and SPP criteria are not necessarily indicators for the real efficacy of CPP policy [22,23]. Some approaches can be found in the literature to measuring the effect of SPP. Cerutti et al. (2018) provided a prospective approach [24]. They used a life cycle assessment (LCA)-based method to assess the potential effect of different SPP practices to support the selection of the most beneficial practice. Retrospective analyses are provided by Larsen and Herwitch (2010) [25], who showed how the carbon footprint of procurement by governments can be calculated based on spending data and environmental extended input-output tables, and by Alvarez and Rubio (2015), who provided an approach to comparing the carbon footprint of two years for a region as a basis to steer SPP [26]. The problem with the latter two approaches is that a change in the procurement footprint is not necessarily caused by SPP or CPP. It can have a range of different causes, such as a change in demands with regard to amounts or quality, a change in production processes, or a change in energy efficiency. In order to analyze the efficacy of CPP, insight is required

for what has been procured and what would have been procured without CPP. Therefore, in this paper, we add to the existing studies by introducing and applying an approach for retrospective analyses of the effect of CPP on the scale of a nation using a mixed-method approach. A mixed method uses and integrates both qualitative and quantitative data in one study [27].

The main aim of this paper is to evaluate whether and to what extent the application of CPP leads to reduced environmental impacts and material use, with the Netherlands as a case study. The results of this paper are meant to feed into national assessments and policies. Furthermore, the results can forward generically applicable CPP implementation practices and contribute to the development of indicators for CPP monitoring. To reach this aim, an existing sample-based mixed-method approach to analyze the implementation and actual effect of SPP was adjusted to the focus of CPP and was applied to seven product groups in the Netherlands. The scope of the study was delineated to impact greenhouse gas emissions (GHG) and material use. The outcomes of this study allow for generalized conclusions which are relevant to improving CPP.

Notice that in this study, the word impact is used to indicate the environmental consequences of purchased goods in their whole life cycle, and the word effect is used to indicate the efficacy of CPP in reducing the impact of purchased goods.

## 2. Materials and Methods

The practical implementation of CPP and its impacts were explored using the method developed by Zijp et al. (2018) [23]. The method is characterized by starting from a macro-level viewpoint, followed by an assessment at the micro level for a sample of selected tenders and then finalized by extrapolation to obtain results on the macro level again. The eight steps within this approach are summarized in Figure 1 and further described below.

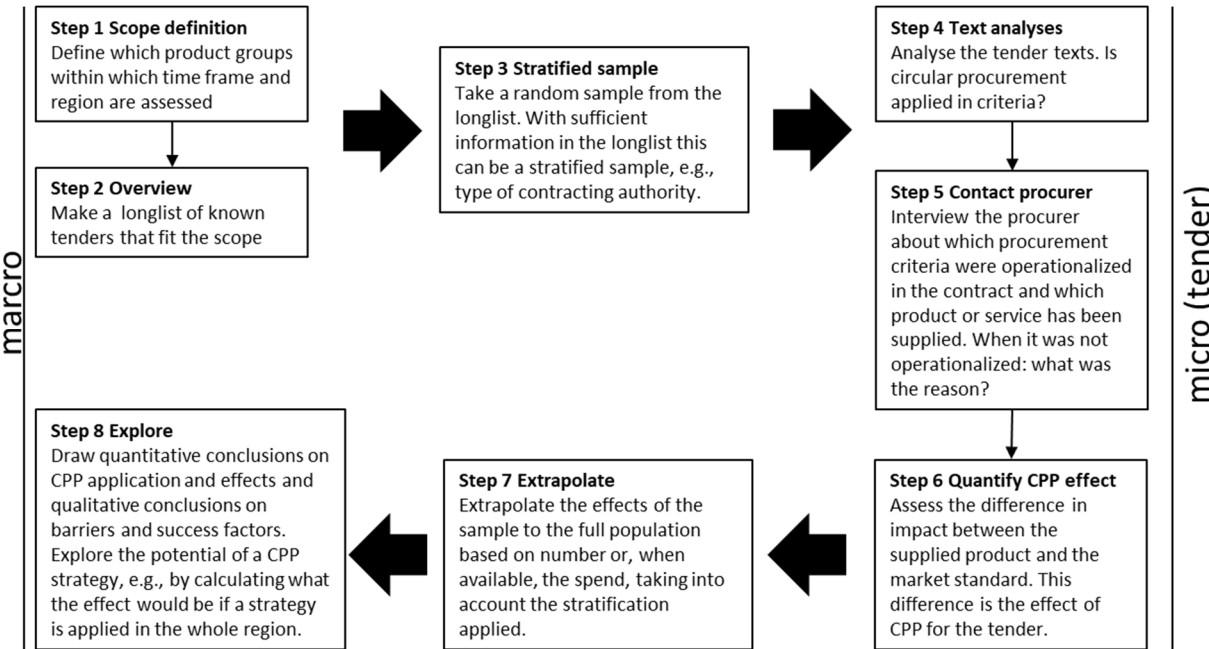

**Figure 1.** Stepwise approach to plan and assess the application and effects of circular public procurement for a nation or region (CPP, based on Zijp et al. (2020) [28]). Effects at individual procurements are extrapolated to insights at the macro level for the whole nation or region.

Step 1 was scope definition. This approach was applied to public procurement in the years 2017–2018. Public procurement was defined as the procurement of goods and services by all governmental bodies in the Netherlands both national and subnational. This included, for example, water boards, schools and universities. Seven product groups

were selected for analysis: road construction, furniture, cars, ICT hardware, solar panels, buildings and work wear. This selection was based on expert judgment of a combination of criteria: (1) material intensity, (2) diversity between product groups, (3) purchasing power of the government and (4) (direct or indirect) relation to the energy transition, as judged by expert pre-scanning. The expert judgement was performed by 10 experts from 7 different institutions of policy and research as well as consultants.

Step 2 involved an overview. Public authorities in the Netherlands are obliged to publish their national and European tenders on the e-procurement platform TenderNed [29]. This platform's database was used to make longlists of tenders in the years 2017–2018. Based on Common Procurement Vocabulary (CPV) codes and screening criteria (i.e., to avoid doubles or void items), a longlist of operationalized procurements per product group was derived. Details on the tenders were collated, such as the contracting authority, description of the required product or service and the expected duration of the contract. Information on the expected expenditure per tender was corrected manually to uniform budget units. In cases of missing data, the gaps were filled with estimated median expenditure values per product group. The longlist for the selection of product groups contained 998 tenders in total.

Step 3 was stratified sampling. Samples were taken randomly but in the same national-to-subnational ratio as the longlist. This was performed because tenders from the national government are most often larger in expenditure than tenders from decentralized governmental bodies, and according to Rosell (2021) [30], they are more likely to include CPP criteria than the smaller ones, although other research shows that some circularity criteria can only be met in a small amount, such as refurbished equipment [31]. A total of 72 tenders was sampled with 10 per product group, except for the buildings product group, for which 12 samples were taken because of its large heterogeneity in CPP measures.

Step 4 was text analysis. The tender texts were scored on hints towards circularity strategies using a double screening process, with reconciliation made by two experts. The experts followed a review protocol, proceeding manually through the key procurement documents and being supported by a search engine to highlight potentially relevant texts. The list of words used for the text mining can be found in Supplementary Materials S1, Table S1. Ideally, this step would be fully performed using text mining, allowing us to analyze all 998 tenders in the longlist. However, we found too many false positives and false negatives to use it for this purpose. The results of the text analyses were structured using the framework of the nine circularity strategies (the so-called R-strategies) as defined by van Buren et al. (2016) [32] (Supplementary Materials S1, Table S2).

Step 5 involved contacting the procurer. Of each tender in the sample, the contracting authority was interviewed by the researchers. The interviews were held with the contract manager responsible for the contract or the general sustainability manager. The following topics were discussed: (1) Did the researchers draw correct conclusions from the application of CPP criteria in the tender? (2) To what extent did the supplied product or service answer the circular procurement criteria asked for? Finally, (3) what are observations of or lessons learned by the procurers in the application of CPP? The specific list of questions raised during the interview depended on the product group and the criteria used in the tenders.

Step 6 was to quantify the CPP effect tender. From the definitions of CPP found in the literature, it becomes clear that CPP should contribute to the transition of CE (e.g., close loops [11] or extend product use [15]). This means CPP is effective when it results in the supply of a product or service which is more circular than what would have been supplied without CPP. Therefore, the effect of circular procurement (*ECP*) of a product or service *i* is defined, for every impact category *j*, as

$$ECP_{i,j,t} = ICP_{i,j,t} - IP_{i,j,t} \tag{1}$$

where $ICP_{i,j,t}$ is the estimated impact of the product or service *i* that is purchased using criteria or requirements which steer toward the optimal and effective (re)use of raw materials and products in impact category *j* over an agreed period of time *t* (life span of the product

or duration of service contract). Impact category *j* could, for example, be greenhouse gas emissions or raw material use. $IP_{i,j,t}$ is the expected impact of a product or service that would have likely been supplied with regular procurement, which is also referred to here as the market standard. Hence, the effect of circular procurement is not the difference between the old and new situations for the contracting authority but the difference that circular procurement makes compared with regular procurement. This is also in line with the perceived goal of the broader SPP: it should stimulate the market for production in a way that is increasingly less impactful, in line with the stimulus embedded in the transitional term Sustainable Development Goals and not the statically defined endpoint of the sustainability goals. The consequence of this definition is that the effect of circular procurement strategies changes over time with a changing market standard (Figure 2).

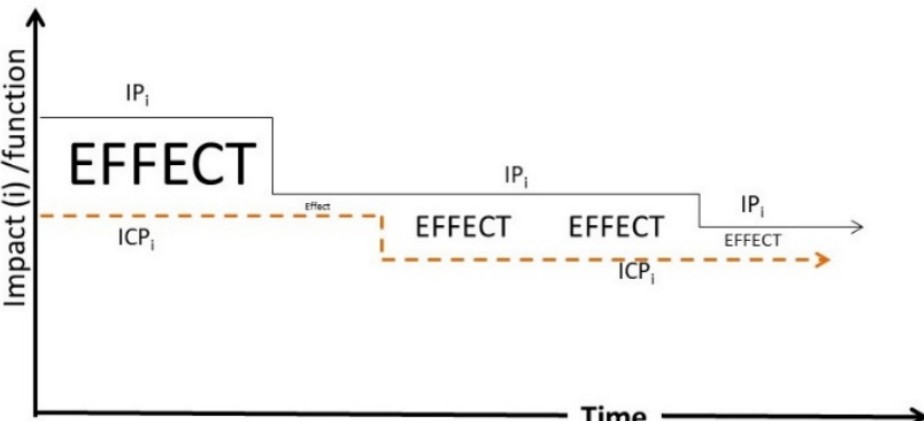

**Figure 2.** Visualization of the definition of the effect of circular public procurement (CPP) as the difference between the impact of regular procurement ($IP_i$) and circular procurement ($ICP_i$). With a changing market standard, the effect of the same CPP strategy becomes smaller. Effective CPP thus requires a continuous challenge to the market in order to show an effect and substantiate continued sustainable development contributions. Derived from [28].

The market standard impact of procurement (*IP*) was derived using the following information: (1) regulations and law, (2) sector-specific reports on market characteristics and developments, such as the Dutch electricity mix, and (3) specific information gathered from different suppliers and procurers. The effect of CPP per tender was assessed using information from the contracting authority or supplier on the product or service, such as the number of products, the materials used and the percentage of recycled content, and existing life cycle inventory data from different sources, such as the EcoInvent Database v.3.5 (Ecoinvent, Zurich, Switzerland [33]). A life cycle perspective was applied in order to include emissions embedded in the materials used and thus allow for identifying possible trade-offs between life cycle stages (extraction, production, use and disposal).

Steps 7 and 8 were extrapolation and exploration. The results of the sample were extrapolated in two ways and for two different means. First, based on the ratio of expenditure between the sample and the longlist, an indication was derived for the effect of CPP in the Netherlands for each product group under investigation. The outcomes were analyzed to create a policy evaluation of past procurement effects. Secondly, cases of successful CPP implementation (with effect) were used to explore "what if" the CPP criteria would have been applied to all governmental purchases in the longlist of the specific product or service. The outcomes were analyzed to derive a policy evaluation of what effect could potentially be expected from CPP as a contribution to CE and climate goals.

The indicators for CPP implementation used for monitoring the implementation of SPP have often been reported as the percentage of tenders in which SPP has been applied thus far [34]. Based on the result of the analysis, four different variants and expansions of this indicator and their result were explored:

(a)  The percentage or number of tenders with criteria that directly ask for circular strategies (reuse, recycling, etc.);
(b)  The percentage or number of tenders with criteria that trigger circular strategies;
(c)  The percentage or number of tenders with criteria that trigger more ambitious circular strategies than the present market standard;
(d)  The percentage or number of tenders with criteria that successfully trigger more ambitious circular strategies than the present market standard.

## 3. Results

### 3.1. Text Analysis

Analysis of the tender texts showed 62 applications of CPP criteria in 31 of the 72 tenders (Figure 3). In total, 24 different user-defined procurement criteria were distinguished. Thus, CPP was applied in 43% of the analyzed tenders, and it seemed that unique criteria were being used often without much repetition. Circularity-related procurement criteria occurred in absolute numbers the least for cars (1 criterion in 1 tender) and the most for furniture (26 criteria applied in 6 tenders) and ICT hardware (10 criteria applied in 8 tenders). Furthermore, analysis of the applied criteria showed that they were mostly related to (Figure 3) (1) repair (R4, 22 applications), applied mainly in tenders on ICT hardware and furniture, (2) longer life span (R2, 13 applications), applied mainly in tenders on furniture, and (3) recycling after use (R8b, 16 applications), applied mainly in tenders on solar panels. Criteria on the refuse (R0), rethink (R1), remanufacture (R6) and repurpose (R7) strategies were not found. For refuse, this was to be expected, because this strategy does not lead to a tender and hence is not found in e-procurement databases. The other strategies can, in theory, be stimulated via procurement criteria, but that was not found in this study.

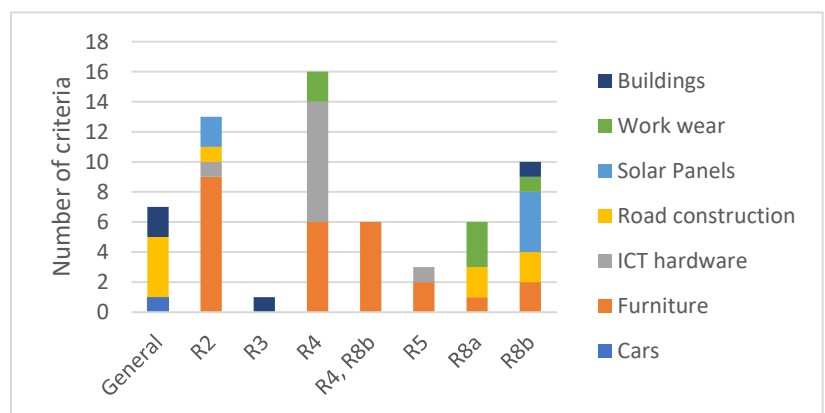

**Figure 3.** Number of R-strategies found in tenders on the seven product groups (R2 = longer life span, R3 = reuse, R4 = repair, R5 = refurbish, R8a = recycled content in product and R8b = fit for recycling after use). Six criteria applications contributed both to R4 and R8b, and this is therefore shown as a separate bar to avoid counting twice. These criteria focused on steering toward easier disassembling of parts, which is beneficial for both repair (R4) and recycling (R8b).

### 3.2. Effect Estimation Based on Interviews

We found the following from the 62 applied CPP criteria:

- Twenty were not clearly deviating from common practice and thus comparable to the market standard. This was the case with, for example, ICT hardware, where requirements regarding availability of spare parts were defined in a very general way, as is common practice. It also occurred with the collection and recycling of solar panels after use, which does not deviate from the common practice described in the WEEE directive [35]. Without more specific requirements for recycling, no extra benefits are to be expected.

- Seven were not operationalized in the contracting phase. Examples included criteria on the recyclability of furniture and workwear after use, which were not operationalized in the contract with the supplier nor secured in the organization of the contracting authority.
- For 21 applications, the ambition and level of operationalization remained unclear. For example, in the case of ICT, the tendering organizations did not administrate how often the option to order refurbished ICT devices within the contract was used. Additionally, no data were gathered to evaluate if the longer guarantee period indeed resulted in longer use of the devices. In the case of furniture, it remained unclear if and how a criterion on easier disassembling was operationalized. Furthermore, agreements on maintenance and repair were not monitored. Data on the type and number of repairs were not gathered for either the reference or the new situation. This type of administration is necessary to assess if CPP application leads to any impact or should be adjusted.
- Fourteen applications were both more ambitious than the market standard and operationalized specifically.

An analysis per the applied criteria found in the sample of 72 tenders can be found in Supplementary Materials S1, Table S1.

Presenting the results on the level of tenders instead of the applied criteria shows the following (Figure 4):

- A total of 43% of the tenders contained one or more CPP criteria;
- Of this, the larger part (38%) used criteria directly related to circularity, and 5% used criteria that triggered circularity indirectly;
- Between 22 and 25% of the tenders contained one or more CPP criteria that were more ambitious than the market standard;
- Between 15 and 21% of the tenders contained one or more CPP criteria that were more ambitious than the market and were also operationalized specifically.

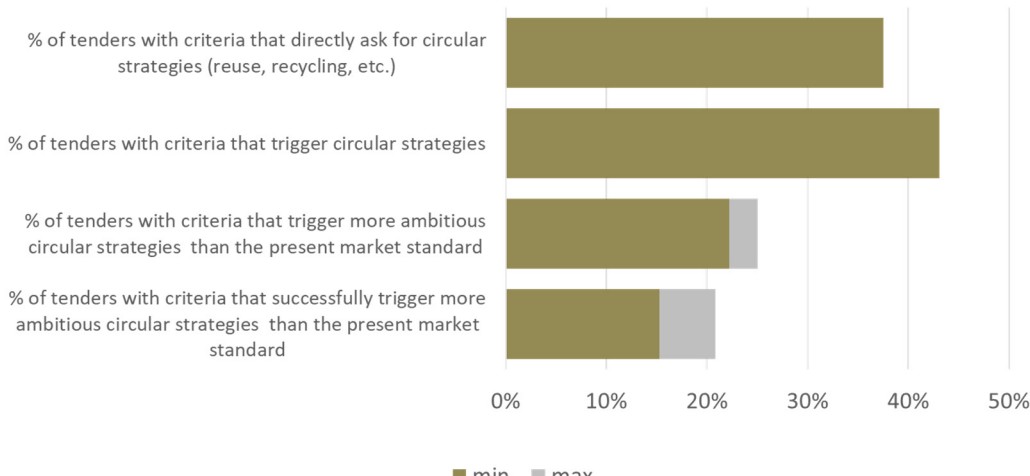

**Figure 4.** Different indicators for CPP implementation provided different scores. The first bar is the percent of tenders with criteria directly relating to circularity (e.g., on recycling and longer lifetime). The second bar is the percent of tenders with criteria that trigger circularity (including, for example, criteria that stimulate reduced carbon footprints). The third bar is the percent of tenders with CPP criteria that are more ambitious than the market standard. The fourth bar is the percent of tenders with CPP criteria that are both more ambitious than the market standard and also operationalized. The light gray bars in the latter two bars reflect the tenders that included criteria for which it remained unclear if the CPP criteria were more ambitious than the market standard or if they were operationalized.

The results show that the score of CPP implementation depended on the definition of what the CPP criteria were. Procurement criteria can trigger circular strategies without explicitly referencing circular strategies such as reuse or recycling. For example, criteria on reducing the carbon footprint of products can and do trigger the development and sale of products with more recycled content, longer life spans and more reuse opportunities. In the sample, three tenders that asked for a $CO_2$ performance ladder certificate [36] led to the supply of products with longer lifetimes and higher fractions of recycled content. Hence, when analyzing CPP implementation and its effects, it is important to realize that a CE is a means to reach environmental and socioeconomic goals, and criteria that focus on these goals can in fact trigger the purchase of more circular products and hence could be considered CPP. For monitoring the implementation of CPP, this means a choice has to be made for including these criteria or not. In particular, the monitoring of CPP implementation based on the results of search engines in the gray and peer-reviewed literature, such as [37], is likely to underestimate CPP application in practice when search terms solely focus on typical circularity vocabulary. Here, 38% of the tenders under consideration contained criteria with direct references to circularity (recycled content, longer life span, etc.). When including criteria that can trigger circular strategies, 43% of the tenders had CPP criteria.

The ranges in Figure 4 illustrate the difference of the outcomes with or without the tenders for which ambition or operationalization remained unclear.

The results show that applications of current CPP criteria are not an indicator for the implementation success of CPP. Up to two thirds of the tenders with CPP criteria were shown to be ineffective in triggering contracting the buying of the CPP-targeted products.

### 3.3. Calculated Effects

The actual effects could be estimated for 10 of the 14 CPP criteria applications, in effect using information on the operationalization and existing LCA data. In the other four cases, the effects could not be estimated. This was due to a lack of data on product design (e.g., percentage of recycled content) and the number of products supplied within the contract. For example, for ICT hardware, one tender set a requirement on the supply of refurbished smartphones during the contract phase when requested by the customer. The contracting authority, however, did not hold accounts on the number of refurbished phones supplied during the contract, and therefore, an effect could not be estimated. The CPP criteria that led to a measurable effect in present day practice in the Netherlands were the following:

- A longer lifetime than the average product (furniture and road construction);
- Refurbishment (furniture);
- Higher recycled contents than the average product (workwear and road construction);
- Sustainability scores or certificates (road construction) which lead to products with longer lifetimes and higher recycled contents.

The results of the effect calculations are presented below per sample and after extrapolation to the whole longlist as indications of the total environmental benefits of CPP for these product groups.

For furniture, 5 of the 10 tenders requested and were supplied furniture with a life span of 15 (2 tenders) or 20 (3 tenders) years instead of the average 10-year guarantee (the market standard). Information on the amounts of furniture supplied within the contracts were retrieved from four of the five contracting authorities. We assumed there would be 1% repair per 10 years. The effect was estimated using Equation (1), where *ICP* was defined as the impact of the requested amount of furniture and 2% repair and *IP* was defined as the impact of 1.5 or 2 times the amount of furniture and 1% repair. LCA data for this comparison were based on the product category rules defined within the project INSIDE/INSIDE (DGBC, [38]). In total, the estimated impact of the use phase of furniture was a saving of 520 tons of $CO_2$eq. emissions and 100 tons of materials used for these 4 tenders (Supplementary Materials S2) [28]. Furthermore, 2 out of the 10 tenders focused on refurbishing furniture during the contract period. Information on the amounts of

furniture refurbished within the first 2 years of the contract could be retrieved from one of the two contracting authorities. Again, the calculations were based on reference profiles provided by INSIDE/INSIDE ([39]), defining the use of furniture for 10 years as a functional unit. The estimated effect for the tender was savings of 110 tons of $CO_2$eq. and 30 tons of material (details and references can be found in Supplementary Materials S2). Together the measures in the sample, this resulted in savings of 630 tons of $CO_2$eq. and 130 tons of materials, which is extrapolated to the longlist of 2960 tons of $CO_2$eq. and 590 tons of materials.

For road construction, 4 of the 10 tenders requested a high score on the certification scheme's $CO_2$ performance ladder [36]. The ladder has several steps which require actions with regard to insight, reduction of energy use, transparency and participation in activities to green the supply chain the applicant is involved in. In all four tenders, a supplier was selected with a high-level certificate (level 5). In three of the four tenders, the supplier proposed circularity measures, of which some were selected by the contracting authorities. In one case, this resulted in a longer life expectancy of the asphalt. Calculations of the impact were based on the Dutch National Milieu Database (NMD) [40] and provided by the supplier. In a second tender, asphalt with 30% recycled content in the top layer was applied, resulting in estimated savings of 469 tons of $CO_2$eq. and 7560 tons of materials. The third tender also used this higher content of recycled content, but the work under this tender was not finished yet. In total, the samples saved 870 tons of $CO_2$eq. and 11,460 tons of materials, which is extrapolated to the full longlist as 23,840 tons of $CO_2$eq. and 291,580 tons of materials saved (details and references can be found in Supplementary Materials S2).

For workwear, 3 out of the 10 tenders requested recycled content. For two of the three tenders, data could be retrieved from the contracting authority on the amount of clothing purchased during the contract phase. Information on the general composition of material per type of clothing was used to translate the amount of clothing into kilograms of fiber. LCA data from various sources were used to estimate the difference between clothing with and without recycled content. One tender resulted in the supply of workwear with recycled polyester instead of virgin polyester, while the other tender resulted in workwear with recycled polyamide contents. The estimated effect of these two tenders was savings of 32 tons of $CO_2$eq. and 7 tons of materials (Supplementary Materials S2). Extrapolation of these results to the longlist of tenders in 2017–2018 results in an estimated potential of 380 tons of $CO_2$eq. and 82 tons of materials saved (details and references can be found in Supplementary Materials S2).

*3.4. Exploration of Measures*

Finally, the potential effects of promising CPP criteria were explored for four product groups.

Regarding furniture, when all tenders in 2017–2018 would have applied a longer life time of 20 instead of 10 years and refurbished 10% of their furniture instead of buying new furniture, this was estimated to result in savings of 53,000 tons of $CO_2$eq and 9300 tons of materials (Supplementary Materials S2) [28].

For buildings, an example of potential measures that could be stimulated by CPP is increasing the circularity of concrete applied in new buildings. Due to the production process of concrete and the amount of concrete generally used in new buildings, it contributed largely to the material and carbon footprint of the buildings. One measure could be to replace 20% of the sand and gravel with concrete granulate. This saves material use (sand and gravel) but requires more energy than regular concrete because of the required breaking of old concrete into granulate. Another measure could be to replace CEM I cement (>95% Portland cement) with CEM III cement (up to 90% slags from blast furnaces). When these two measures are applied to the estimated yearly amount of newly built governmental buildings in the Netherlands (EIB 2019), an estimated 292 tons of $CO_2$eq. and 4.2 ktons of material (sand and gravel) could be saved (Supplementary Materials S2).

In road construction, based on the estimated yearly amount of asphalt used in the Netherlands, extrapolation of the effects found in the sample indicated 24 kilotons of $CO_2$eq. and 300 kilotons of materials could be realized (Supplementary Materials S2). When the two identified measures of longer life and higher recycled content were applied in all road construction and maintenance work in the Netherlands, this could result in estimated GHG savings of 0.6 Mtons of $CO_2$eq. and 11.5 Mtons of material (Supplementary Materials S2) [28]. It must be emphasized that this is a potential estimation if 50% recycled content can be reached. This potential was based on the road area to be replaced and not the available supply of secondary material.

For workwear, cotton is the most commonly used fiber in the Netherlands. Post-consumer recycling of cotton is challenging, however, mainly because wearing of the fabric and the mechanical recycling process itself result in shorter fibers, which are harder to spin (fine) yarn from. Other challenges are sorting collected used fabric based on fiber type, the efficient removal of zippers and buttons and recycling fiber blends. Finding technical solutions to these challenges could be stimulated by governmental organizations by actively deploying CPP criteria on recycled content. The interviewees estimated the maximum post-consumer recycled content of cotton in workwear which can presently be provided by the market to be 10%. When all governmental tenders on workwear in 2017–2018 would have used this criterium, the estimated saved GHG emissions would have been 600 tons of $CO_2$eq. and 190 tons of materials (Supplementary Materials S2).

## 4. Discussion and Conclusions

The analyses, using the sample-based mixed-method approach, showed that application of CPP in 2017–2018 in the Netherlands led to effects in one third of its applications. Combining qualitative and quantitative approaches in the mixed-method research design allowed us to (1) narrow down the need for quantitative data to the samples where an effect was expected based on the qualitative data and (2) analyze the main reasons for two thirds of the CPP criteria applications not resulting in a climate benefit and reduced material consumption. The main reasons were that (1) in many tenders, the applied criteria lacked ambition, and (2) in some tenders, ambitious criteria were not operationalized.

The first reason—tenders were frequently comparable or even less ambitious than product market standards—might be found in the organizational change required to successfully implement sustainable procurement and thus also CPP. It requires changing existing organizational routines, which takes time [41]. It seems that in many situations, applying CPP criteria is not yet backed up with the organizational change necessary for ambitious application. What might play a role, and what takes time to change, is that according to Rolfstam (2012) [42], previous procurement policies have led to procurers developing risk-averse behavior. However, in order to launch customers and procurement that stimulate the transition toward a circular economy, room for experiments and innovation is required. In a study in the context of realizing an innovative circular bridge, Lenderink et al. showed the importance of the government in making innovations possible by, among other methods, limiting financial risks, explicitly managing the risk (e.g., by ensuring a fallback option) and clearly splitting the developing and realization phases of the project [43]. Hence, in order to successfully implement CPP with impact, public authorities should not walk away from risks but facilitate management of the risks for all involved parties. Another role the government can play in the implementation of impactful CPP has been proposed by different authors, such as Kannan in her study on drivers for sustainable procurement in Denmark [16] and Zaidi et al. in their study on factors that resist sustainable procurement in Pakistan [20]. They concluded that two main drivers of sustainable procurement are governmental regulation and legislation and pressure from stakeholders such as customers. Kannan suggests that governments set minimum mandatory requirements. The results of our study add to this advice by showing that when setting these minimum requirements, this should be performed in such a way that it stimulates more ambitious procurement than the current market standard. The preparation of mini-

mum requirements can already trigger the market to change, resulting in a new market standard before or shortly after the CPP criteria are operationalized. Hence, developing mandatory minimum requirements that stay ambitious enough to trigger the market is an important field of investigation.

The second reason is that ambitious tenders and offers were not implemented, showing that the organizational change required not only focuses on the procurement department of the contracting authorities but also involves the whole organization. The success of CPP does not depend on a single activity in the procurement procedure, but it can increase or decrease in its latter phases as well [44]. Exemplary are, for example, the tenders that ask for products (in this study furniture and workwear) to be easily recyclable, but in the contracting phase, no activities were defined yet to make sure the products would indeed be recycled after disposal. Because circular products can be used in a linear way, circular procurement should include processes that secure the operationalization of the R-strategies. This is a point of attention, especially for the R-strategies that focus on the end of the life cycle (i.e., reuse, longer use and recycling).

This conclusion on the necessity of whole organization implementation aligns with studies from different geographical regions in both developed and developing countries. These studies point toward the importance of top management support [18,45] in the context of the United States and China, leadership in sustainability [19] in the context of the United Kingdom and environmental commitment "urging environmental awareness in every area of the business" in the context of Germany [17]. Various authors showed that the role of change agents seems to be crucial to realize this holistic implementation, such as Grandia (2015) [46] in the context of the Netherlands.

The results of this study also show that the percentage of tenders with CPP- or SPP-related criteria is not necessarily a good indicator for the national implementation of SPP and CPP; that is, successful implementation depends not only on the use of CPP or SPP criteria but also on their ambition and their implementation. The indicators proposed in this study provide a more detailed estimation of implementation success and provide insights that can help to design measures for improvement in CPP practices. However, for the latter part especially, the actual estimation of effect was shown to be challenging due to the lack of data on what had been supplied or lack of insight in the current reference situation. Hence, the approach points to important data requirements needed for effect-based policy making. In situations that lacked data for quantification of the effect, the qualitative assessment (if the effect was expected or not) proved valuable as well. Although the results do not answer the question of to which extent the policy instrument CPP contributes to reaching policy goals, it does provide insight on the level of implementation success and insights into the reasons for (a lack of) effectiveness.

The applied approach proved to be versatile (i.e., applicable in the different contexts of the different product groups). It was developed for the situation in the Netherlands, where there is an overview on tenders thanks to e-procurement, but no central administration exists for contracts and what products have actually been supplied. E-procurement covers only the procurement via tenders. Smaller purchases are thus a blind spot in e-procurement-based approaches, such as those applied in this study. On the other hand, Rosell (2021) [30] showed that the application of CPP is more likely to occur in larger than in smaller tenders. The sample-taking phase is essential in situations such as the one in the Netherlands, were there is no central administration on what has been contracted and supplied. The sample size was determined by the budget of the research. More statistically sound samples can be taken when there is more information on the variability in CPP application and its potential effects. To make this possible, automatic search engines (that also cover the tender annexes) are being explored, but until now, too many false negatives and positives appeared for this strategy to be useful for tender selection.

The product groups were chosen based on, among other criteria, material intensity and thus their expected relevance for closing material loops. The selection of product groups, and hence the results of this study, are not meant to be representative for all procurement of

the Dutch government. One of the criteria for selection was relation to the energy transition. This is of specific importance because the energy transition requires technology that can have trade-offs between GHG emission reductions and increased use of materials such as, among others, rare earth metals [47]. Here, the energy and circularity transition should go hand in hand, and thus co-designed CPP criteria should be applied in the procurement processes of these technologies.

The results of the CPP effect assessment depend largely on the definition of the market standard. Market standards change over time (Figure 2). The available information to determine this market standard varies largely, from well-documented and discussed (such as the average energy mix or type of cars leased in a specified period) to hardly a sound basis to define a market standard (such as the current repair practice of ICT devices used by the government). Steering toward and monitoring the evaluation of such cases of more repairs would require better insight into the current situation. Because the argumentation on what the market standard should be varies per product group, transparency about the market standard used is important, and the consequences of different approaches to defining the market standard should be further investigated.

The applied approach is able to analyze the effect of CPP measures that are described in tender documents. As such, the approach especially covers the product- and technology-oriented CPP measures. Of the four ways in which contracting authorities can implement CPP, according to Alhola et al. (2021) [15], the first three (the procurement of better-quality products in circular terms, the procurement of new circular products and the use of business concepts that support the CE) are likely to be covered by the proposed monitoring approach, as long as the CPP technical and process measures are mentioned in the tender documents. For the latter one, investment in circular ecosystems, a direct relation between a tender and the impact will be harder to prove. How to quantitatively analyze the possible impacts of that strategy requires further research. The same applies to the R-strategy of buying less (refuse). This will have an advantageous effect, but as explained in Section 3.1, it is not covered by the proposed approach because it is not part of the tender documents that form the basis for our approach. Similarly, in procurement for buildings or road construction, the design precedes the tender phase. While in the design phase a lot of circular and sustainable decisions can be made [43], these design choices are not always visible in the tender documents.

In conclusion, despite the current limited availability of data and the need to further investigate the way to estimate the market standard, this study adds to the current literature by providing a unique insight into the approaches to and effects of CPP in the Netherlands. The results underline the relevance of these insights, as two thirds of the tenders using CPP criteria were deemed ineffective. The mixed-method approach provided insights on the reasons for the lack of effect, which aligned with suggestions from the recent literature from the organizational change point of view. Insights on environmental effects, uncertain effects and the (likely) absence of effects are key to eventually effectively contributing to the transition toward a CE and to assist in reaching the UN SDGs. Just like how a CE is a transition, monitoring the CE requires a transition as well. A common language needs to be developed (what do we want to measure and against which baseline), and behavior needs to be changed (allowing the storage and gathering of data on the required level). This manuscript contributes to both transitions through more effective monitoring that makes more effective CPP possible.

**Supplementary Materials:** The following supporting information can be downloaded at https://www.mdpi.com/article/10.3390/su141610271/s1. S1: List of words used for text analysis and overview of CPP criteria found in the samples. S2: References to data used for LCA calculation.

**Author Contributions:** M.Z., E.D. and L.P. designed the research framework; E.D., M.H., M.B., J.T., A.D.K., A.H. and E.D.V. gathered and analyzed the data; M.Z. drafted the first draft of the manuscript, and all the other authors contributed by drafting reviewing and editing different parts of the texts. All authors have read and agreed to the published version of the manuscript.

**Funding:** This study was performed as part of a national research program aimed at monitoring and evaluating the progress of the transition to a circular economy in the Netherlands led by the Netherlands Environmental Assessment Agency (PBL). The research was funded by PBL grant number: E/121069/01/AA.

**Institutional Review Board Statement:** Not applicable.

**Informed Consent Statement:** Not applicable.

**Data Availability Statement:** Links to relevant data can be found in Supplementary Materials S2.

**Acknowledgments:** We specially thank Anne Gerdien Prins (PBL) for reviewing the early drafts of this manuscript.

**Conflicts of Interest:** The authors declare no conflict of interest.

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
