# Peer review of "Measuring the Effect of Circular Public Procurement on Government’s Environmental Impact"

_sustainability, doi:10.3390/su141610271_

Round 1

Reviewer 1 Report

The research presented is highly valuable as it addresses a blind spot of the emerging literature about CPP.

The question of CPP's ability to contribute to sustainability is highly discussed in the literature but we definitely lack valuable empirical data. The research realized here is original, robust and very useful.

I just have two minor remarks:

1- The literature review is short. The definition of CPP could be more discussed using works from scholars who have tried to specify this new topic.

For instance, cheng et al. (2018) is cited in the references but not mentionned in the literature review. Same thing for Sönnichsen & Clement (2020). You could check also works by Alhola et al. (2019).

2- The link between the results and conclusions can be improved. The results are clearly exposed for the different products categories showing for instance the level of CPP effect, but I do no see clearly how you reach to the conclusion that some of the failures in implementation of CPP come from organizational issues. It could be interesting to provide some examples to illustrate and support this point.

Author Response

Dear reviewer,

thank you for your kind words about the value of the manuscript. Based on your comments we were able to improve the manuscript. Please find our response  to your, and the other reviewers', comments in the manuscript in the attachment. All comments contain a reference to the line numbers at which we made changes based on your suggestions.

Kind regards, on behalf of all authors,

Michiel C. Zijp

Reviewer 2 Report

This paper is very interesting and well structured. It attaches to an very important issue-- CPP/GPP.

Especially, the discussion on the effect of implementation of CPP indeed exist research gap. I think this paper can fill that gap in the perspective of EU and contract analysis.

I highly recommended publish this paper after some minor revision.

1) The author needs to examine many details throughout the paper, There are some words that don't take serious. For example. Line 18 the attr. "CCP". the CCP means Chinese Communist Party, I don't think the authors want to express this meaning.

2) It could be better if the author can read more international perspective papers focus on CPP/GPP, it could improve the impact of this paper and expand the scope of discussion. I suggest 3 papers to follow.

[1] https://doi.org/10.1016/j.pursup.2020.100622

[2] https://doi.org/10.1080/09640568.2021.1978060

Finally, I think this paper can enrich the discussion on the CPP/GPP topic and its very worth to be published. Thank you and best regards.

Author Response

(The authors gave the same response as above.)

Author Response

(The authors gave the same response as above.)

Round 2

Reviewer 3 Report

The revision has added value to the paper.